# Physicochemical Studies on the Surface of Polyamide 6.6 Fabrics Functionalized by DBD Plasmas Operated at Atmospheric and Sub-Atmospheric Pressures

**DOI:** 10.3390/polym12092128

**Published:** 2020-09-18

**Authors:** Larissa Nascimento, Fernando Gasi, Richard Landers, Argemiro da Silva Sobrinho, Eduardo Aragão, Mariana Fraga, Gilberto Petraconi, William Chiappim, Rodrigo Pessoa

**Affiliations:** 1Laboratório de Plasmas e Processos, Instituto Tecnológico de Aeronáutica (ITA), São José dos Campos 12228-900, Brazil; argemiro@ita.br (A.d.S.S.); petra@ita.br (G.P.); 2Centro de Engenharia, Modelagem e Ciências Sociais Aplicadas, Universidade Federal do ABC (UFABC), São Bernardo do Campo 09210-170, Brazil; fernando.gasi@ufabc.edu.br; 3Instituto de Física Gleb Wataghin (IFGW), Universidade Estadual Paulista (Unicamp), Campinas 13083-859, Brazil; landers@ifi.unicamp.br; 4Campus Integrado de Manufatura e Tecnologias, SENAI Cimatec, Salvador 41650-010, Brazil; eduardo.aragao@fieb.org.br; 5Instituto de Ciência e Tecnologia (ICT), Universidade Federal de São Paulo (Unifesp), São José dos Campos 12231-280, Brazil; mafraga@ieee.org; 6i3N, Departamento de Física, Universidade de Aveiro, Campus Santiago, Aveiro 3810-193, Portugal

**Keywords:** atmospheric plasma, sub-atmospheric plasma, dielectric barrier discharge, polyamide 6.6, surface modification, wettability, whiteness

## Abstract

This work proposes the use of a dielectric barrier discharge (DBD) reactor operating at atmospheric pressure (AP) using air and sub-atmospheric pressure (SAP) using air or argon to treat polyamide 6.6 (PA6.6) fabrics. Here, plasma dosages corresponding to 37.5 kW·min·m^−2^ for AP and 7.5 kW·min·m^−2^ for SAP in air or argon were used. The hydrophilicity aging effect property of untreated and DBD-treated PA6.6 samples was evaluated from the apparent contact angle. The surface changes in physical microstructure were studied by field emission scanning electron microscopy (FE-SEM). To prove the changes in chemical functional groups in the fibers, Fourier transform infrared spectroscopy (FTIR) was used, and the change in surface bonds was evaluated by energy dispersive X-ray spectroscopy (EDS) and X-ray photoelectron spectroscopy (XPS). In addition, the whiteness effect was investigated by the color spectrophotometry (Datacolor) technique. The results showed that the increase in surface roughness by the SAP DBD treatment contributed to a decrease in and maintenance of the hydrophilicity of PA6.6 fabrics for longer. The SAP DBD in air treatment promoted an enhancement of the aging effect with a low plasma dosage (5-fold reduction compared with AP DBD treatment). Finally, the SAP DBD treatment using argon functionalizes the fabric surface more efficiently than DBD treatments in air.

## 1. Introduction

Nylon 6.6, also known as polyamide 6.6 (PA6.6), is a copolyamide prepared from a mixture of a hexamethylenediamine and adipic acid [1]. PA6.6 compounds have excellent physical–chemical, mechanical, and electrical properties. It worth highlighting the melting temperature of 250 °C, the tensile strength of 81 MPa, the low weight, the excellent machinability, the dimensional stability, the dielectric constant of 3.6, and the breakdown voltage of 16 kV/mm [1,2]. These characteristics, associated with lower manufacturing costs, make PA6.6 an important thermoplastic used in molded components used in the automotive industry [3,4,5], filtration applications [1,6,7], and numerous engineering applications [8]. However, in the past two decades, PA6.6 has become one of the prominent polyamide fibers in the textile industry, used for the manufacture of industrial textiles, yarns, carpets, and clothing [9,10]. Despite the excellent aforementioned properties, due to their nature, PA6.6 fibers have weaknesses in terms of hydrophobicity and low surface energy, which reduce clothing comfort, coloring, and adhesion properties for large-scale industrial applications [10,11]. These issues generate high economic and environmental costs for the textile industry, mainly related to chemical pretreatment that consumes large amounts of chemical compounds, water, and energy [12]. Chemical treatment is used as a method of the functionalization of PA6.6 fabrics, i.e., it creates new functionalities for textiles, in many cases enabling new applications [9,13].

As an alternative to the conventional chemical functionalization of fabrics, a treatment based on cold plasma, also called non-thermal plasma, emerged [14,15,16,17,18,19]. This dry method has the advantages of being environmentally friendly, worker friendly, operating at atmospheric or sub-atmospheric pressure, and can modify the surface of the textiles without affecting their bulk properties, in addition to being suitable for most heat-sensitive polymeric textile materials [20,21,22,23,24]. The types of plasmas most used on textiles are used at atmospheric pressure, namely (i) plasma jet (PJ), (ii) glow discharge (GD), (iii) corona discharge (CD), and (iv) dielectric barrier discharge (DBD) [10,11,12,14,15,16,19,21,22,23,24]. Among them, DBD technology appears as a promising method for the functionalization of polymeric textiles. A significant benefit of DBD over other types of gas discharges is the higher electron density induced by micro-discharges that are generated by a large number of small current filaments that cross both dielectric layers (over one or both electrodes). This characterizes DBD technology and prevents gas breakdown or spark through current limitation [25], helping to improve the functionalization of textiles by the high electron density that impinge on the surface of the textile evenly [26].

In this article, we report on the functionalization of commercial PA6.6 fabric by a homemade DBD reactor operating at atmospheric pressure (AP) or sub-atmospheric pressure (SAP). Due to the lack of studies comparing DBD operated in AP and SAP using atmospheric air and argon as a gas source for plasma generation, this work focused on the assessment of the functionalization behavior of PA6.6 fabrics carried out in: (i) AP using air, (ii) SAP using air, and (iii) SAP using argon. The understanding of these two parameters (gas pressure and plasma chemistry) is crucial for the textile industry. Moreover, the vacuum treatment allows for better control of the process atmosphere, allowing for a more in-depth study of the effects of each type of gas on the structural and chemical changes of the treated surface. In this study, we used the following techniques: (a) spectrophotometric analysis using Datacolor to measure the whiteness, (b) apparent contact angle analysis to measure the wettability, (c) field emission scanning electron microscopy (FE-SEM) to evaluate surface morphology, and (d) Fourier transform infrared spectroscopy (FTIR) and X-ray photoelectron spectroscopy (XPS) to study chemical structure of the bulk and surface of the substrates, respectively. To ensure better measurement accuracy, each sample has a duplicate that has been treated by DBD and investigated under the same conditions. Thus, we obtain an average for all measurements.

## 2. Materials and Methods 

### 2.1. Polyamide Sample Preparation and Plasma Treatment

Pristine half-knitted fabric composed of 92% polyamide 6.6 (PA6.6, Rhodia Solvay Group, Santo André, Brazil) and 8% elastane (Lycra brand, Invista, São Paulo, Brazil) was used in this work. This knitted fabric was produced in fine machinery (38 needles per inch) with a surface mass of 180 g/m^2^ [27]. The fabrics were cut into 10 × 8 cm^2^ pieces and then washed with a 1% non-ionic detergent solution at room temperature for 30 min and rinsed with water for 10 min to avoid contamination. This procedure was performed before the DBD treatment. Figure 1 provides the FE-SEM images of the pristine PA6.6 sample. The methodology used was based on the division of work into four sets of samples: (i) pristine PA6.6 (control); (ii) sample 1 (S1) (PA6.6 treated at AP); (iii) sample 2 (S2) (PA6.6 treated at SAP with air plasma); and sample 3 (S3) (PA6.6 treated at SAP with argon plasma). It is important to note that each set has duplicate samples that were processed and analyzed under the same conditions for better measurement accuracy.

Figure 2 shows the schematic diagram of the homemade DBD reactor used to functionalize the PA6.6 fabrics. The DBD plasma is generated between the polarized and grounded electrodes. This last electrode (made of an aluminum cylinder 165 mm high and 135 mm in diameter) was covered with a 5 mm thick dielectric layer made of silicone material. The polarized electrode (made of a copper tube 115 mm high and 6 mm in diameter) can be covered with silicone depending on the operating pressure of the DBD. In this work, the electrodes were separated by 2 mm in order to generate a uniform discharge along the electrode axis (see Figure 3). In addition, the polarized electrode had micro-holes that acted as a gas injection system, inducing micro-discharges that passed through the dielectric layers and generated a large number of small current filaments between the electrodes. 

To generate the DBD plasma, we used a homemade power supply composed of a signal generator (Minipa, model MFG-4221, São Paulo, Brazil), a power amplifier (Profisson 16.2, Rio Grande do Sul, Brazil), and a high voltage transformer. The copper electrode was connected to the high voltage output of the transformer, providing a peak-to-peak voltage of 15 kV at a frequency of 23 kHz in burst mode of the signal generator (with the number of oscillations *N* = 20 and a repetition period *T*_r_ = 5 ms) with dissipation of electrical energy in the discharge volume being of the order of 6.30 W for AP and 1.26 W for SAP in air or argon. The electrical characterization of the plasma was performed by measuring the transferred power using the Lissajous figure method. For this, we used a 10 nF capacitor, a resistance of 85 Ω, a high voltage probe 1:1000 (Tektronix, model P6015A, Beaverton, OR, USA), a self-adjusting probe 1:10 (Agilent Technologies Inc., model N2863B, Santa Clara, CA, USA), and a 300 MHz digital oscilloscope (Agilent Technologies Inc., model InfiniiVision DSO5032A, Santa Clara, CA, USA). Data acquisition was done by placing the capacitor and resistor in series with the grounded electrode, this was in order to measure the load transferred on the capacitor. The current was obtained by measuring the voltage of the resistor with the self-adjusting probe, the applied voltage was monitored by the high voltage probe.

The process chamber was coupled to the grounded electrode and to a rotating sample movement system. All samples treated with DBD plasma were subjected to a rotation of 0.5 rpm for 10 treatment cycles with the gas flow controlled by rotameters and mass flow controllers. For 10 treatment cycles, the plasma dosage corresponded to 37.5 kW·min·m^−2^ for AP and 7.5 kW·min·m^−2^ for SAP in air or argon [28]. As shown in Figure 3a under the AP condition, the powered electrode is covered by a 2 mm thick silicone to prevent sparking, and note that DBD plasma is uniform along the PA6.6 substrate. In Figure 3b,c, the powered electrode is bare and we use a Pyrex glass bell (365 mm high and 394 mm in diameter) in order to reduce the gas pressure to 30 torr for both gas types (air or argon) using a mechanical vacuum pump (Edwards Brazil, E2M18, Barueri, Brazil).

After the plasma treatment, the sample was kept in vacuum-sealed plastic bags for further characterization and comparison with the untreated (control) sample.

### 2.2. Polyamide Surface Characterizations

The contact angle measurements were performed at room temperature with deionized water. For this, a ramé-hart goniometer (Ramé-Hart Instrument co., model 590, F4 series, Succasunna, NJ, USA) was used. Here, we performed the evaluations using droplets of (2.5 ± 0.1) µL deposited by a syringe on PA6.6 samples (control, S1, S2, and S3). The contact angle value was measured as the average of four consecutive measurements, being carried out on non-moistened points in the sample. DROPimage Advanced software (Ramé-Hart Instrument co., Succasunna, NJ, USA) was used for data acquisition. This characterization was carried out over 8 days, starting immediately after the DBD treatment and followed up day by day. 

The analysis of the degree of whiteness of the fabrics was performed with Datacolor SpectraVision 650 equipment (Datacolor Inc., Lawrenceville, NJ, USA). This characterization was performed right after treatment by DBD, and the chromaticity index was obtained through software and verified by the CIELAB color space system. This analysis allowed us to quantify the spectral distribution of the color by the reflectance of the fabric through the following parameters: luminosity, clarity, hue, saturation, and chromaticity using the *L**, *a**, *b** and *C* index obtained in Cartesian coordinates. 

The images of the surface morphology of the PA6.6 fabrics before and after exposure to DBD plasma were taken with a field emission scanning electron microscope (FE-SEM), model Tescan Mira 3 FEG (TESCAN Brno, s.r.o., Kohoutovice, Czech Republic), operated at 5 kV. Additionally, the energy dispersive X-ray spectroscopy (EDS) technique was used to explore chemical modifications in PA6.6 fabrics. 

To investigate the changes in chemical functional groups in the fibers, an FTIR spectrometer coupled with attenuated total reflectance (ATR) accessory model PerkinElmer Frontier (PerkinElmer, São Paulo, Brazil) was used. A resolution of 2 cm^−1^ was used. This analysis was performed using a blank ATR cell (PerkinElmer, São Paulo, Brazil) as the background to record all spectra [29]. X-ray photoelectron spectroscopy (XPS) was also used to analyze the surface of the PA6.6 fabrics both untreated and treated by DBD plasma. The instrument used was the VSW HA-100 Spherical Analyzer (VSW Scientific Instruments LTD., Manchester, UK) operating with 44 eV energy. An X-ray source of Al-Kα was used with an exciting photon energy of 1486.6 eV with a spatial resolution of less than 5 µm, a compliance angle of 90°, and maximum vacuum of 6 × 10^−8^ mbar. XPSpeak software (Version 4.1, Dr. Raymond Kwok, Hong Kong, China) was used to analyze the data corresponding to the reference signal C1s signal with a binding energy of approximately 285 eV. All XPS spectrum fitting used the Gaussian–Lorentzian (GL) curve method to analyze the data. Using the photoionization cross section together with the electron escape depth, the photoelectron peak areas that were used to calculate the relative surface compositions of all samples were normalized. Finally, atomic compositions were estimated through the integrated peak areas of C1s, O1s, N1s, Si2p, and Si2s.

## 3. Results and Discussion

### 3.1. Surface Morphological Analysis and Chemical Analysis of the Inner Part of the Fibers

Figure 4a–i show the FE-SEM images of samples S1, S2, and S3 for three different image amplifications, which show a change in the topography of the fabrics treated by air plasma in AP and SAP, and argon plasma in SAP. This change in the morphology of the samples treated with plasma was compared based on the control sample (shown in Figure 1). Three different magnitudes were used to verify the characteristic interlacing of the sample, the fiber layout, and the changes caused by the plasma on its surface. Figure 4g shows a greater ablation in the surface of the fibers for the sample treated in AP plasma (S1). In Figure 4g, a sub-micron size non-uniform etching induced by DBD plasma on the surface structure of the PA6.6 fiber is shown. As shown in the literature, this etching is caused by highly energetic and reactive species from AP plasma [26] that increase the surface roughness and change the hydrophobic nature of PA6.6 to hydrophilic [30,31,32]. It is worth mentioning that, in addition to the etching effect, there may be a break in the PA6.6 molecule chains, generating new functional groups or reorganizing the existing polymer groups [33].

Figure 4h shows the FE-SEM image of the PA6.6 sample treated by SAP plasma (S2), where a lower ablation was observed with better uniformity compared to S1. Despite using the same plasma chemistry (air source), this behavior can be attributed to the lower applied power and gas pressure that decreased the energy of electrons and free radicals, reducing ablation and interactions with the following reactive molecules: N_2_^+^, N_4_^+^, N^+^, O_2_^+^, H_2_O^+^, O_2_^−^, and O^−^, which caused less formation of new functional groups on the surface of polymers [33]. Due to the high complexity of the interaction between plasma and the surface of the PA6.6 fabric, it is not trivial to differentiate by FE-SEM images if the more uniform etching on S2 is related to the reduction of the chemical structure of the surface caused by plasma or is related to material removed through the impinging of plasma species on the surface. On the other hand, Figure 4i shows a sub-micron-sized non-uniform etching induced by DBD plasma at SAP on the surface structure of PA6.6 fiber using argon as the plasma source. In comparison with sample 2, it was observed that sample 3 presented a more effective roughness in the same 30 torr of pressure. This behavior is due to the contribution of the physical etching effect promoted by argon plasma [34,35].

The FE-SEM image in Figure 5a shows a decrease in the diameter of the outermost fibers after treatment with argon plasma, of which the innermost PA6.6 fiber had a diameter of 11.12 µm and the fiber exposed to bombardment with plasma species had a diameter shrinkage of 8.05 µm. This fact highlights the more superficial action of plasma on textiles, even when operated at low pressure.

This shrinkage in the diameter of the PA6.6 fiber also occurred for S1 and S2, where air was used in the treatment with DBD plasma (not shown). This behavior leads us to suppose that, in addition to the superficial changes, the plasma treatment is also responsible for changes in the internal part of the fibers, corroborating their shrinkage. The control sample, S1, S2, and S3 were characterized by the FTIR technique in order to explain this observed phenomenology. As shown in Figure 5b, the FTIR spectra of the samples treated in the three process parameters compared to the control sample are comparatively similar, with slight variation in peaks related to the characteristics of PA6.6.

As the reach depth varies between 0.5 µm (4000 cm^−1^) and 5.0 µm (400 cm^−1^), the FTIR technique becomes an excellent tool for understanding and checking for possible chemical changes in the inner layers of PA6.6 fibers. Figure 5b shows the bands related to the internal bonds of pristine PA6.6, a behavior that is repeated for all DBD-treated samples. The first bands associated with PA6.6 are the N–H bonds at 3290, 1460, and 750 cm^−1^ attributed to elongation, deformation, and flexion vibrations [36]. Bands related to the CH bonds of the CH_2_ (stretching vibrations) and CH_3_ groups (bending vibration) in terminal groups of the PA6.6 chains appear at 2860 and 2932 cm^−1^ [37]. The stretching vibrations related to the C=O bonds can be observed around 1732 cm^−1^ [38]. At 1634, 1535, and 1373 cm^−1^, there is stretching, asymmetric deformation, and bending related to NH amide groups, respectively [39]. Between 1000–1300 cm^−1^, bands corresponding to the N-mono-substituted amide group of the PA6.6 chain of the CN bonds of the amide were observed [40]. The bands located around 1140 cm^−1^ can be attributed to the symmetrical bending vibration of CO–CH combined with the twist of CH_2_ [41]. Bands at 932 and 679 cm^−1^ were associated with the stretching and bending vibrations of the C–C connections, and the group at 573 cm^−1^ may be attributed to the O=C–N flexion [42]. The bands at 936 and 1140 cm^−1^ were associated with the crystalline and amorphous structures of PA6.6, respectively [43].

Therefore, as observed in the FTIR spectra, there is a slight change in the transmittance intensity of the peaks mentioned above. This behavior shows that the shrinkage of the fiber is related to the surface modification and not to changes in the chemical bonds of the inner part of the fiber. This statement is in resonance with the literature, which demonstrates that the influence of photons on polyamide textiles can reach a depth of several tens of nanometers. For other particles, this interaction is limited below 10 nm [44,45,46].

### 3.2. Wettability and Whiteness Analysis of the Fabrics

The wettability/hydrophilicity of PA6.6 fabrics was analyzed by measuring the apparent contact angle. All analyses were performed at two dry points on the fabric pieces and repeated on duplicate samples, where the error was ± 0.5° for all samples. Figure 6 shows the apparent contact angle data collected during the first seconds of water drop contact with the PA6.6 surface and over eight days. The initial value of the contact angle of the control sample was (115.0 ± 0.5)°, which demonstrates the hydrophobic nature of the PA6.6 fabric. After the application of DBD plasma in AP using air plasma (S1), the contact angle obtained was 0°, showing the instant adsorption of the drop of water. According to Oliveira et al. [47], this behavior occurred with PA6.6 fabrics exposed to plasma dosages of 2.0 to 3.5 kW·min·m^−2^. The DBD plasma was generated in a semi-industrial prototype machine at atmospheric pressure. They suggested that the PA6.6 surface changes were activated by the applied plasma, which increased the water adsorption rate for the range of plasma dosages investigated. For our case, the plasma dosage was 37.5 kW·min·m^−2^ in an atmospheric pressure condition.

In addition, in the present work, we also investigated the SAP DBD treatment with a plasma dosage of 7.5 kW·min·m^−2^ for air plasma, allowing us to obtain an apparent contact angle of 0°, thus showing the instantaneous adsorption of the water droplet. When argon plasma in SAP (S3) was used, immediate adsorption was also observed. The average adsorption time of the water droplet for the sample control was (65 ± 7) s, validating the hydrophobic nature of the PA6.6 fiber. After DBD plasma treatment in AP and SAP for both gas sources (air or argon), a decrease in the average adsorption time to (3.0 ± 0.2) s for S1, (4.5 ± 0.3) s for S2, and (2.8 ± 0.2) s for S3 was observed. This behavior indicates a functional surface change in all cases after plasma treatment, changing the nature of PA6.6 from hydrophobic to hydrophilic. Therefore, immediately after the DBD treatment regardless of the gas source used (air or argon) or gas pressure (AP or SAP), hydrophilicity is achieved in the investigated plasma dosages. Similar wettability results were reported for natural and synthetic materials [48,49,50,51], with the DBD plasma being essential for this change, due to the increase in surface energy, superficial cleaning, and alteration of the superficial chemical bonds that improve the hydrophilicity of textiles [18,26,29,47,50,51].

In addition to the initial apparent contact angle, the effect of aging on surface modification over time was studied. The aging study was performed in PA6.6 fabric before (control sample) and after exposure to DBD plasma (samples S1, S2, and S3) using apparent contact angle measurements. The wettability of each sample was measured day by day over 8 days, to validate stability effects generated by non-thermal plasma treatment. Figure 6 shows the impact of the AP and SAP plasma dosages on PA6.6 fabric through the temporal study of the apparent contact angle of a drop of deionized water. Pristine PA6.6 samples obtained a contact angle of 115 to 120° over eight days of aging. The sample S1, in the first four days after treatment, presented hydrophilicity (θ < 90°), with a critical change from the fourth day in apparent contact angle. In the first two days, total water adsorption (0°) occurred and S1 become hydrophobic on the fourth day. The sample S2, in the first three days after treatment, showed hydrophilicity behavior (θ < 90°), with a critical change in apparent contact angle ((65 ± 4)°, (77 ± 5)°, and (87 ± 3)°). On the fourth day, an apparent contact angle with a value of (105 ± 7)° became hydrophobic (θ > 90°) and reached the value of the untreated sample (120°) on the eighth day. These results corroborate the FE-SEM images, which showed greater ablation/etching in the S1 sample treated at AP. Consequently, the increase in the surface roughness contributed to altering and maintaining the hydrophilicity of PA6.6 fabrics for a longer time [31,32]. In comparison, the S2 sample had a greater increase in the effect of aging due to the low plasma dosage (reduced by five times). Therefore, for the same plasma chemistry (air source), SAP functionalizes the fabric surface more efficiently than AP. 

The S3 sample was treated with argon plasma at SAP, and in the first five days after the treatment it presented hydrophilicity and became hydrophobic on the sixth day. This result shows better functionalization using argon compared to air in AP or SAP, suggesting that the chemistry of argon plasma is more efficient than air plasma. Karahan and Özdogan attributed the improved functionalization and the more significant roughness and, subsequently, better wettability to the significant etching effect of the argon plasma [34]. 

The loss of hydrophilicity obtained during the plasma treatment (aging effect) is documented in the literature, being dependent on storage, temperature, and environmental conditions [52,53]. The dynamic behavior of the chemical bonds on the polymer surface, the contamination of the surface, and dissociation of the gas molecules by plasma, generating reactive radicals, are the dominant hypotheses for the aging effect caused by non-thermal plasma treatment [54,55].

The parameters *a** and *b** in Cartesian coordinates were used to define the degree of whiteness of the plasma-treated fabric [56]. Positive values of *a** represent the range of redness, while the negative values represent the range of green. On the other hand, the positive *b** values are linked to the yellow spectrum and the negative *b** values are associated with the blue spectrum. Figure 7 presents the graph of the parameters *a** and *b** in Cartesian coordinates of the samples treated with DBD plasma (S1, S2, and S3), taking as a reference the primary color sample PA6.6. These color measurement parameters were calculated using the equations presented by the Commission Internationale de l’Eclairage (CIE) [56] and Pozo-Antonio et al. [57]. It can be seen that the parameter *b** for the surface treated with DBD in air at AP (S1 sample) exhibited an increase in the slope towards yellow compared to the control sample. This behavior is related to the more significant roughness shown in Figure 4g, which changes the degree of reflectance and, consequently, the brightness of the sample reflected in the color spectrophotometer [58]. Samples S2 and S3 showed a slight increase in parameter *b** compared to the control sample, probably due to less roughness.

In comparison with the degree of yellowness of the sample S1, the sample S2 has a value equivalent to 21.1% (0.73) of S1 (3.46), and S3 has a value equivalent to 24.3% (0.84) of S1. The parameter *a** is approximately uniform for the three plasma-treated samples with values of−0.36, −0.49, and −0.41, respectively, for samples S1, S2, and S3, showing a slight tendency to greenness. The results mentioned earlier showed that samples S2 and S3 have parameters of *a** and *b** that tend towards the whiteness spectrum.

Therefore, the results of wettability and whiteness demonstrated that the DBD argon plasma in SAP treatment was responsible for more significant parameters, i.e., the aging effect was retarded for one day in comparison to the sample functionalized by air plasma at AP and retarded for two days in comparison to DBD air plasma in SAP, and, in relation to whiteness, low values for *a** and *b** indices were obtained, showing a higher degree of whiteness.

### 3.3. Surface Chemical Analysis of the Fabrics

To study the degree of chemical modifications on the surface of the substrates with a depth range between 5 and 10 nm, XPS analysis was used. With this technique, it is possible to evaluate the surface oxidation, the atomic percentage, and the number of nanoparticles on the surface of fabrics [10,47]. EDS analysis was used as a comparative technique with an acceleration voltage of 5 kV (depth reaching up to 0.5 μm) to analyze the oxidation and atomic percentage from the surface to the interior of the fibers [59,60,61].

Figure 8 shows the XPS spectra of the untreated and treated samples, and it can be seen that PA6.6 samples treated with DBD plasma using air or argon have the highest O1s peak intensity compared to the control sample. The presence of C, O, N, and Si in the composition of the untreated and DBD-treated fabrics was observed. We consider the sum of C, O, N, and Si to be 100%, i.e., we neglected the ratio of H in the calculations.

As shown in Table 1, there is a decrease in the percentage of carbon after plasma treatment, with the ablation/etching of the substrate responsible for breaking the polymer chains of groups C–H, C–O, C–N, and N–H and the subsequent formation of functional oxygen groups, increasing the atomic O/C ratio [47,62,63]. In all samples after plasma treatment, N (%) decreased and Si (%) remained approximately constant. The data presented in Table 1 show an excellent agreement between the values presented by the EDS (500 nm) and XPS (5–10 nm) analysis, despite the different depth range between the two techniques. The XPS values were estimated through the integrated peak areas of C1s, O1s, N1s, Si2p, and Si2s. The spectra of O1s, N1s, Si2p, and Si2s are not shown in this work.

It is worth highlighting that this increase in the ratio (O+N)/C may be related to the increase in wettability, as demonstrated by the contact angle analysis (Figure 6). Another parameter related to the higher value of the ratio (O+N)/C is the yellowness of sample 1, which, in addition to roughness, may be responsible for this behavior.

In order to investigate the chemical bonding states on the pristine PA6.6 surface, as well as on the plasma-treated surfaces, the core-level XPS spectra of C1s were used and the XPS data were analyzed by the peak split, using Gaussian–Lorentzian (GL) equations to describe the peak energies related to chemical bond groups (Figure 9). This method consists of adjusting a baseline that was used in all spectra. With the help of the GL equations, the spectrum is split into GL curves that describe the different energy diagrams of specific groups. Finally, we fit the experimental data with an envelope, i.e., a sum of all energy spectrum diagrams.

Figure 9a shows the C1s spectrum of the pristine PA6.6 fabric, where the prominent peak at 284.5 eV is related to C=C bonds and the peak at 285.5 eV corresponds to C–N bonds. The other three peaks at 282.6 eV, 286.3 eV, and 287.9 eV are attributed to C–Si, C–O (ether group), and C=O (carbonyl group), respectively [10,64,65,66,67,68]. After the plasma treatment, the peak positions of the C1s spectra did not change, only the relative intensities of the peaks were modified. The specific relative amount of carbon bond groups is shown in Table 2. In the case of the sample S1 (Figure 9b), a decrease in the relative intensities of C=O, C–N, and C–Si with a consequent increase in C=C bonds was observed. Still, a new chemical bond appears with a slight intensity at 288.8 eV, corresponding to COOH bonds. This new functional group reflects the changes in C, N, and O, as shown in Table 1, and this oxidation is related to the hydrophilicity of the S1 sample [10,12,14,17,22,46,47].

On the other hand, when DBD in SAP with air was used (sample S2, Figure 9c), a decrease in the C=C and C–N bonds with an increase in COOH and C–Si bonds was observed. The rise in the relative intensity of the COOH bond is related to the hydrophilicity of the S2 sample [10]. Although there is no evidence, we can suggest that the increase in the relative intensity of the C–Si bond is related to the aging effect on wettability, i.e., in comparison with S1, the sample S2 has retarded aging for one day.

Figure 9d shows the C1s spectrum of the S3 sample, and an increase in the integrated intensity for the C–Si energy spectrum was observed when using argon plasma in SAP. Table 2 shows that the relative quantity of C–Si bonds increased more than three times in relation to the control sample and, consequently, there is a drastic decrease in the relative quantity of C=C bonds. As mentioned earlier, this behavior of surface bonds may be related to the aging effect on wettability, so we can assume that sample S3 reached the longest time in the hydrophilic state due the increase in the relative amount of the C–Si bonds caused by argon plasma treatment. Although the argon plasma has an inert chemistry, it promoted oxidation in the PA6.6 fabric in the same way as the air plasma, as shown in Table 1 and Table 2.

Therefore, both plasma chemistries break the carbon bonds of the original surface and promote the formation of hydroxyl, carboxyl, and carbon double bonds that improve the hydrophilicity of the fabrics [10,47,68]. It is worth mentioning that the removal of impurities on the surface promoted by plasma treatment is responsible for increasing surface energy and, consequently, for increasing hydrophilicity [26].

## 4. Conclusions

Treatment with atmospheric and sub-atmospheric DBD plasma using plasma dosages of 37.5 and 7.5 kW·min·m^−2^, respectively, induced physical and chemical modifications on the surface of commercial PA6.6 fabrics. The FE-SEM images showed superficial changes in the physical microstructure of the fabrics, with a more significant ablation occurring in the S1 sample due to the atmospheric pressure. FTIR analysis showed that shrinkage after plasma treatment did not affect the interior bonds in the deep layer of the fibers. The apparent contact angle measured in the initial moments after treatment with plasma DBD showed the hydrophilicity of all PA6.6 fabrics, regardless of the pressure or the plasma chemistry used. In addition, the analysis of the apparent contact angle carried out day by day over eight days showed that the sub-atmospheric pressure improved the aging effect on the wettability of samples S2 and S3 for one and two days, respectively, compared to sample S1. According to the XPS analysis, this behavior is related to an increase in the relative intensity of the C–Si energy spectrum from 4.51% to 5.63% and 13.41% in samples S2 and S3, respectively. The analysis of EDS and XPS show that an increase in the ratio (O+N)/C of the DBD-treated samples corroborates the higher number of polar groups on the surface that improved the wettability and increased the surface energy. The XPS C1s spectra show peaks at 282.6, 284.5, 285.5, 286.3, 287.9, and 288.8 eV that have been attributed to C–Si, C=C, C–N, C–O, C=O, and COOH bonds. Finally, the Datacolor analysis showed a more significant degree of whiteness for samples S2 and S3 due to the lower roughness caused by DBD treatment under sub-atmospheric pressure. 

## Figures and Tables

**Figure 1 polymers-12-02128-f001:**
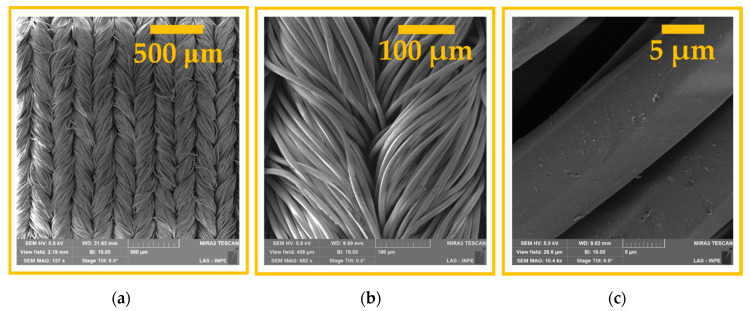
Field emission scanning electron microscopy images of pristine PA6.6 samples with amplifications of: (**a**) 171×; (**b**) 682×; and (**c**) 10,400×.

**Figure 2 polymers-12-02128-f002:**
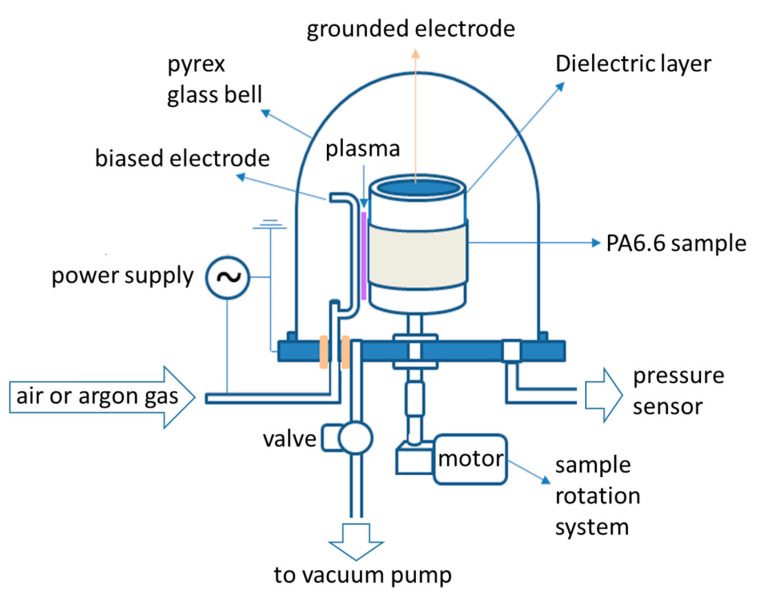
Schematic diagram of the dielectric barrier discharge reactor applied for atmospheric or sub-atmospheric PA6.6 fabric treatment.

**Figure 3 polymers-12-02128-f003:**
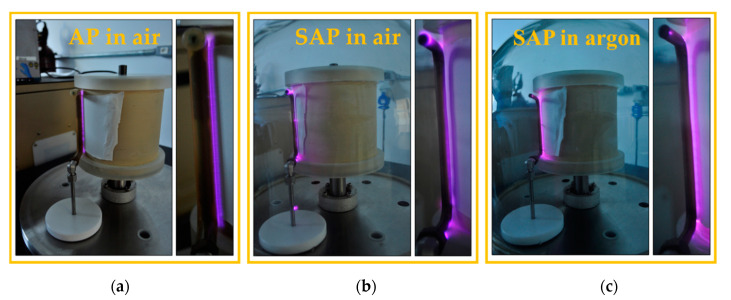
Photographs of the process chamber and dielectric barrier discharge (DBD) plasma operating in: (**a**) atmospheric air; (**b**) sub-atmospheric pressure in air at 30 torr; and (**c**) sub-atmospheric pressure in argon at 30 torr.

**Figure 4 polymers-12-02128-f004:**
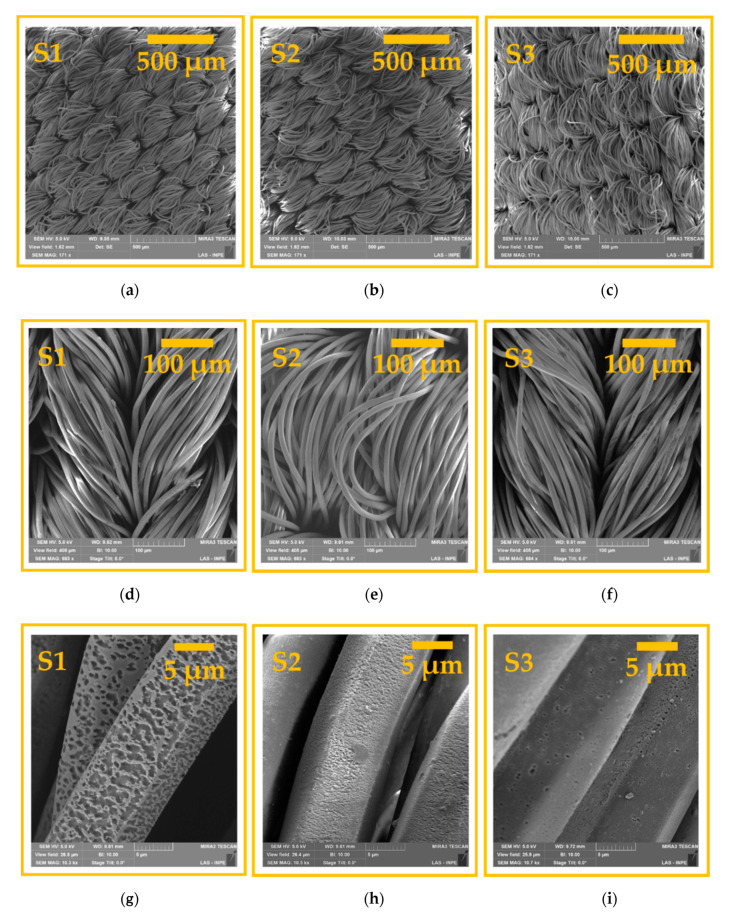
Field emission scanning electron microscopy images of samples S1, S2, and S3 with magnitudes of: (**a**–**c**) 171×; (**d**–**f**) 663×; (**g**–**i**) 10,300×.

**Figure 5 polymers-12-02128-f005:**
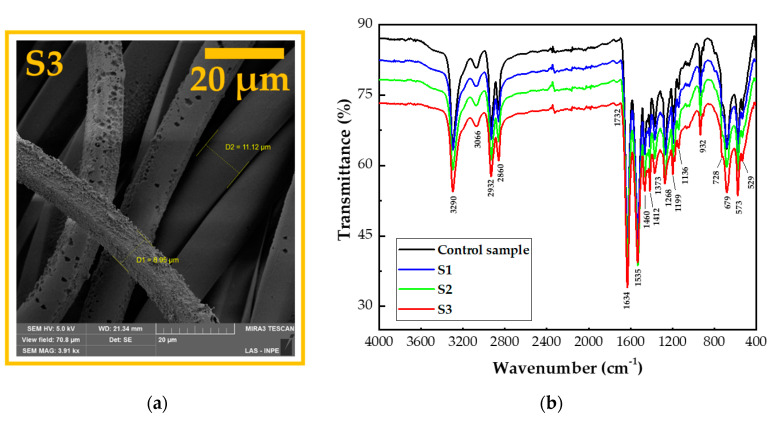
(**a**) Field emission scanning electron microscopy image of S3 fabrics with amplification of 3910× and (**b**) FTIR spectra of the untreated and DBD-treated samples.

**Figure 6 polymers-12-02128-f006:**
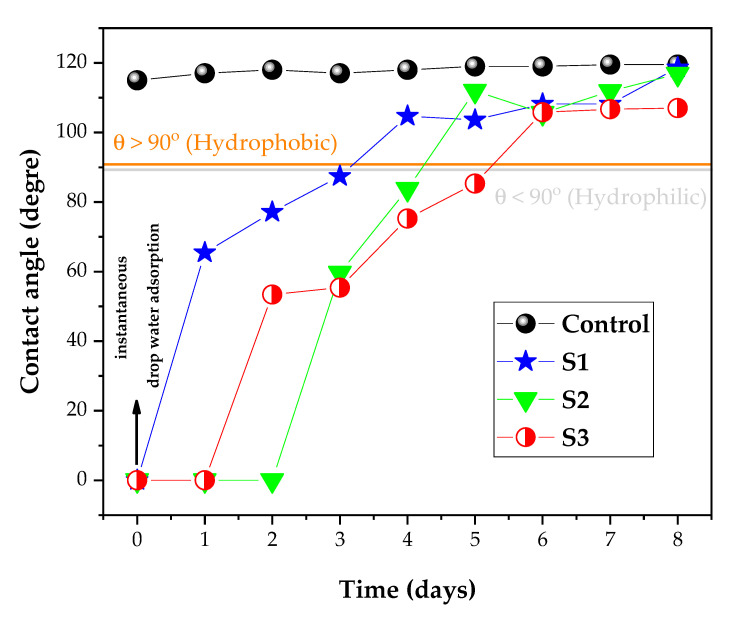
Apparent contact angle of the control sample and samples S1, S2, and S3 measured just after DBD plasma treatment and for eight days after treatment. All analyses were carried out on dry points of the pieces and repeated in duplicate samples, where the error bar is ± 0.5 degrees (not shown in the plot).

**Figure 7 polymers-12-02128-f007:**
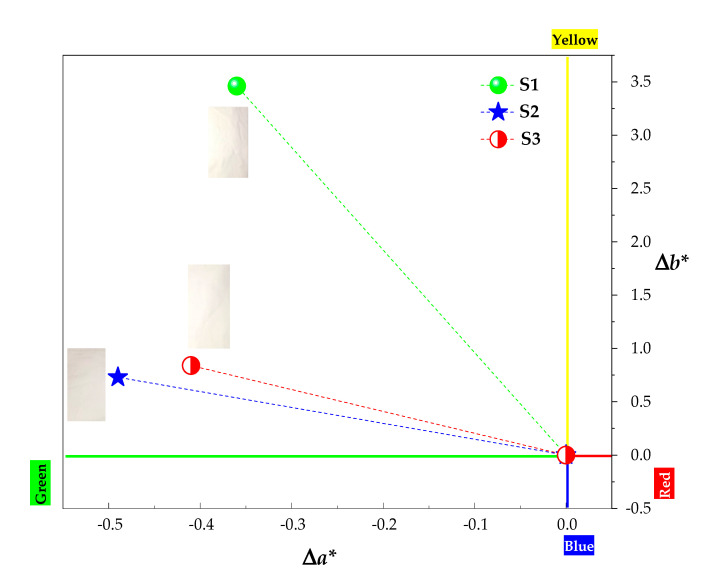
The plot of the parameters *a** and *b** used to verify the degree of whiteness of the samples treated with non-thermal air plasma in atmospheric pressure (AP) (S1) and sub-atmospheric pressure (SAP) (S2), and with non-thermal argon plasma in sub-atmospheric pressure (SAP) (S3). Images of samples S1, S2 and S3 were inserted in the figure to better visualize the degree of whiteness.

**Figure 8 polymers-12-02128-f008:**
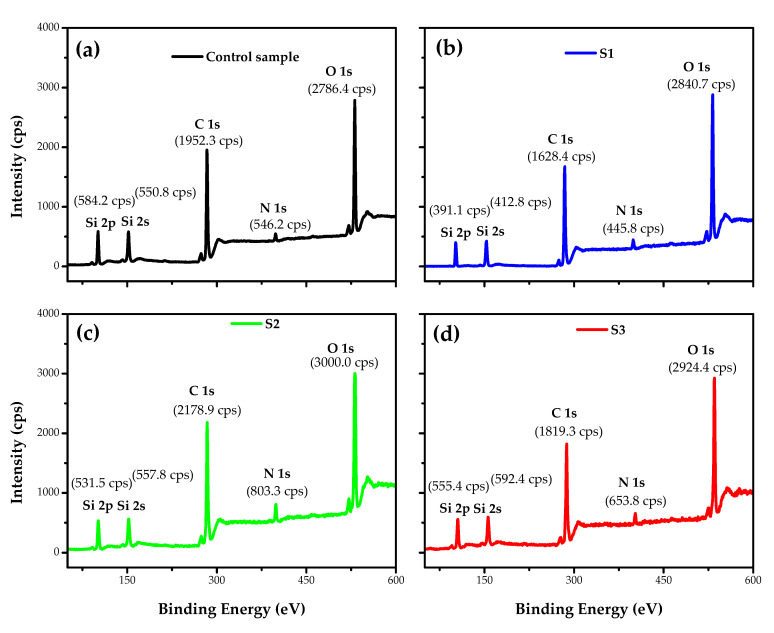
X-ray photoelectron spectroscopy (XPS) spectra of pristine PA6.6 fabric (**a**); DBD-treated in AP (**b**); DBD-treated in SAP using air (**c**); and DBD-treated in SAP using argon (**d**).

**Figure 9 polymers-12-02128-f009:**
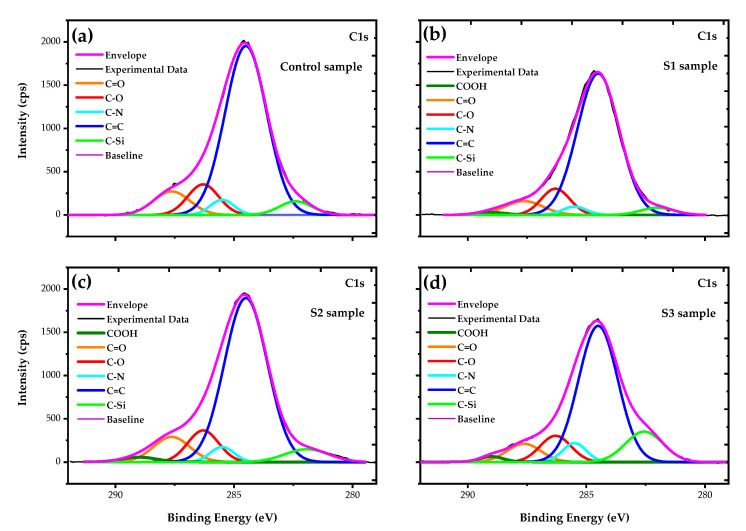
XPS deconvoluted curves of C1s spectra for the control and DBD-treated PA6.6 fibers in AP and SAP: (**a**) control sample with five energy spectrum diagrams at 287.9, 286.3, 285.5, 284.5, and 282.6 eV, respectively, C=O, C–O, C–N, C=C, and C–Si bonds; (**b**) S1, (**c**) S2, and (**d**) S3 samples show the same five energy spectra with different relative intensities, with a new peak at 288.8 eV related to the COOH bond.

**Table 1 polymers-12-02128-t001:** Atomic composition and ratio obtained by energy dispersive X-ray spectroscopy (EDS) and XPS analysis (the error of the results was estimated to be ±5%).

Atomic Composition and Ratio									
Analysis	Samples	C (%)	O (%)	O/C	N (%)	N/C	Si (%)	Si/C	(O+N)/C
**EDS**	Control	46.40	39.40	0.85	6.62	0.14	7.58	0.16	0.99
	S1	33.40	52.80	1.58	3.80	0.13	10.00	0.30	1.69
	S2	35.05	51.60	1.47	4.06	0.12	9.29	0.27	1.58
	S3	37.90	48.80	1.29	3.93	0.10	9.37	0.25	1.39
**XPS**	Control	52.20	33.13	0.63	5.56	0.11	8.09	0.15	0.74
	S1	31.04	55.95	1.80	3.54	0.11	8.05	0.26	1.92
	S2	32.33	54.37	1.68	3.85	0.12	9.06	0.28	1.80
	S3	34.55	51.93	1.50	3.64	0.11	9.88	0.29	1.61

**Table 2 polymers-12-02128-t002:** The relative amount of carbon bond groups shown in the XPS spectra (the error of the results was estimated to be ±5%).

Sample	Carbon Bond Groups (%)					
	COOH	C=O	C–O	C–N	C=C	C–Si
Control	0	8.87	9.80	3.72	73.00	4.61
S1	1.06	6.80	9.69	2.55	76.80	3.10
S2	1.71	9.13	9.76	3.56	70.21	5.63
S3	1.54	7.30	9.55	5.30	62.90	13.41

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
