# Peer review of "Physicochemical Studies on the Surface of Polyamide 6.6 Fabrics Functionalized by DBD Plasmas Operated at Atmospheric and Sub-Atmospheric Pressures"

_polymers, 2020, doi:10.3390/polym12092128_

Round 1

Reviewer 1 Report

The paper "Physico-chemical studies on the surface of polyamide 6.6 fabrics functionalized by DBD plasmas operated at atmospheric and sub-atmospheric pressures" is devoted to the important problem, however, it needs  a very serious revision.

Remarks:

1. In the text:

"The wettability/hydrophilicity of PA6.6 fabrics was analyzed by measuring the dynamic contact angle"

"In addition to the instant contact angle, it was studied the aging effect of the surface modification over time. The aging study was carried out in polyamide fabric before (control sample) and after DBD plasma exposure (samples S1, S2, and S3) using dynamic contact angle measurements. Their wettability was measured day-by-day over eight days, to validate stability effects generated by non-thermal plasma treatment".

Actually the authors did not measure the dynamic contact angle. The dynamic contact angle emerges, when a droplet moves (!) relatively to the substrate, see:  

Voinov O. V. Hydrodynamics of wetting, Fluid Dynamics, 1976, 11, 714-721.

Hoffman R. L. A study of the advancing interface, J. Colloid & Interface Sci.1975, 50, 228-241.

Bormashenko Ed. Physics of wetting. Phenomena and applications of fluids on surfaces, de Gruyter, Berlin, 2017.

Bonn D., Eggers J., Indekeu J., Meunier J., Rolley E. Reviews of Modern Physics 2009, 81, 739-805.

Actually, what was measured by the authors is an "apparent contact angle".

The use of correct scientific wording is extremely important.

2. The authors studied the wetting of the micro-rough surface. It remains unclear to a reader, what kind of wetting regime was observed; namely, was it the Wenzel-like or the Cassie-like (heterogeneous) wetting?

3. An apparent contact angle itself does not exhaust the characterization of wetting of the reported fabric; the contact angle hysteresis necessarily should be reported, see:

Bormashenko Ed. Physics of wetting. Phenomena and applications of fluids on surfaces, de Gruyter, Berlin, 2017. 

4. In the text:

"Instantaneous measured contact angle shows the hydrophily"

The notion of "instantaneous contact angle" is misleading and erroneous; the notion of the "apparent contact angle" should be used.

What is "hydrophily"? This is an obvious typos.

5. In the text:

"Commercial polyamide 6.6 (PA6.6) with silica embedded in its structure needs to be functionalized to improve the reduced hydrophobicity and the low surface energy".

The statement is obscure. What is the "reduced hydrophobicity"? Why is it reduced? Why the surface energy is low? It is low when compared to what material?

Author Response

Manuscript ID: polymers-932848

Title: Physicochemical Studies on the Surface of Polyamide 6.6 Fabrics Functionalized by DBD Plasmas Operated at Atmospheric and Sub-Atmospheric Pressures

Journal: Polymers

Dear Dr. Yulia Yu

We are thankful for your e-mail with the reviewer’s comments. In addition, we are very pleased to know that the most of the reviewers were satisfied with the manuscript and considered it appropriate to be published in this journal after the revisions required. We work hard to answer all questions, as you can see below. Also, the English language was carefully revised along the whole article text.

We would like to thank the reviewers for their comments / suggestions and we hope that this revised version of our manuscript – that takes into consideration some of the reviewers’ suggestions – meets the requirements for publication in the Polymers Journal. Changes in the the manuscript are highlighted in yellow.

Best regards,

The authors

Response to the Reviewer 1 comments and list of changes made to the manuscript:

The paper "Physico-chemical studies on the surface of polyamide 6.6 fabrics functionalized by DBD plasmas operated at atmospheric and sub-atmospheric pressures" is devoted to the important problem, however, it needs  a very serious revision.

Remarks:

  1. In the text:

"The wettability/hydrophilicity of PA6.6 fabrics was analyzed by measuring the dynamic contact angle"

"In addition to the instant contact angle, it was studied the aging effect of the surface modification over time. The aging study was carried out in polyamide fabric before (control sample) and after DBD plasma exposure (samples S1, S2, and S3) using dynamic contact angle measurements. Their wettability was measured day-by-day over eight days, to validate stability effects generated by non-thermal plasma treatment".

Actually the authors did not measure the dynamic contact angle. The dynamic contact angle emerges, when a droplet moves (!) relatively to the substrate, see:  

Voinov O. V. Hydrodynamics of wetting, Fluid Dynamics, 1976, 11, 714-721.

Hoffman R. L. A study of the advancing interface, J. Colloid & Interface Sci.1975, 50, 228-241.

Bormashenko Ed. Physics of wetting. Phenomena and applications of fluids on surfaces, de Gruyter, Berlin, 2017.

Bonn D., Eggers J., Indekeu J., Meunier J., Rolley E. Reviews of Modern Physics 2009, 81, 739-805.

Actually, what was measured by the authors is an "apparent contact angle".

The use of correct scientific wording is extremely important.

Response: We agree with the reviewer. Several corrections were made throughout the text of the article.

  1. The authors studied the wetting of the micro-rough surface. It remains unclear to a reader, what kind of wetting regime was observed; namely, was it the Wenzel-like or the Cassie-like (heterogeneous) wetting?

Response: This discussion is interesting if we consider increasing the contact angle with the days. We believe that Cassie-like wetting can be considered in the discussion of the wetting regime, mainly due to the different surface roughness. However, due to the complexity and smaller contact angles during aging effect, this analysis is not easy and we believe it may be interesting for a future article.

  1. An apparent contact angle itself does not exhaust the characterization of wetting of the reported fabric; the contact angle hysteresis necessarily should be reported, see:

Bormashenko Ed. Physics of wetting. Phenomena and applications of fluids on surfaces, de Gruyter, Berlin, 2017. 

Response: Although the visualization of the movement of the drops, hydrophobicity and water repellency from the perspective of the contact line structure and kinetic barriers to the movement of the contact line is a matter of interest, our equipment does not allow this type of measurement in an automated way, being necessary use other equipment. We intend, in a future work, to carry out such measurement in order to better discuss the effects of wettability on PA6.6 caused by the different types of DBD plasmas used.

  1. In the text:

"Instantaneous measured contact angle shows the hydrophily"

The notion of "instantaneous contact angle" is misleading and erroneous; the notion of the "apparent contact angle" should be used.

What is "hydrophily"? This is an obvious typos.

Response: We agree with the reviewer. Several corrections were made throughout the text of the article in order to correct the typos and some erroneous definitions, specially concern the contact angle results and analysis.

  1. In the text:

"Commercial polyamide 6.6 (PA6.6) with silica embedded in its structure needs to be functionalized to improve the reduced hydrophobicity and the low surface energy".

The statement is obscure. What is the "reduced hydrophobicity"? Why is it reduced? Why the surface energy is low? It is low when compared to what material?

Response: We agree with the reviewer. The phrase "Commercial polyamide 6.6 (PA6.6) with silica embedded in its structure needs to be functionalized to improve the reduced hydrophobicity and the low surface energy" was removed from article abstract and text.  

Reviewer 2 Report

The paper presents results of investigations on polyamide fabrics functionalized with plasma. The obtained results are interesting and worthy to publish in Polymers.

The applied procedure results in promising effects. However, the treatment was performed in laboratory conditions for small samples with dimensions 10 x 8 cm2. Application of this treatment as continuous process in industrial conditions will be a great challenge.

Before publication the paper has to be revised.

Comments:

1/ Generally English has to be improved.

There are unfortunate expressions and misused terms in the text.

For example:

  1. 79 “As a contribution to the literature, in the present paper, we studied the functionalization …”. I think the goal of research is to study a new treatment, not to produce next paper.

l.103 the term “lower” is relative. One can discuss if the magnification of 10 000 belongs to low magnifications. Preferably other term should be used.

  1. 111 The term “amplification” does not fit to microscopic images. Commonly used is the term: “magnification”

l.173 To investigate the chemical bonds of the sample. Do you really studied chemical bonds?

There are many grammatical errors in the text.

For example:

  1. 114-115 … was covered with 5 mm thick dielectric made of silicone material. Probably the word “layer” is missing.
  2. 172 … coupled to investigated the chemical modifications

l.179 The source is Al-Ka radiation source ….

  1. 183 … curve method to analyses the data

There are much more unfortunate expressions, misused terms and grammatical errors. All have to be eliminated.

2/ There is lack of information regards fabric in the text. The reference concerning this point is given. Nevertheless, short characteristic of the fabric used in investigations should be presented.

3/ Results and discussion

The first three sentences in the paragraph (l.190-195) are unnecessary. Two sentences are suitable for the Introduction. One sentence repeats information from the experim

Author Response

Manuscript ID: polymers-932848

Title: Physicochemical Studies on the Surface of Polyamide 6.6 Fabrics Functionalized by DBD Plasmas Operated at Atmospheric and Sub-Atmospheric Pressures

Journal: Polymers

Dear Dr. Yulia Yu

We are thankful for your e-mail with the reviewer’s comments. In addition, we are very pleased to know that the most of the reviewers were satisfied with the manuscript and considered it appropriate to be published in this journal after the revisions required. We work hard to answer all questions, as you can see below. Also, the English language was carefully revised along the whole article text.

We would like to thank the reviewers for their comments / suggestions and we hope that this revised version of our manuscript – that takes into consideration some of the reviewers’ suggestions – meets the requirements for publication in the Polymers Journal. Changes in the the manuscript are highlighted in yellow.

Best regards,

The authors

Response to the Reviewer 2 comments:

The paper presents results of investigations on polyamide fabrics functionalized with plasma. The obtained results are interesting and worthy to publish in Polymers.

The applied procedure results in promising effects. However, the treatment was performed in laboratory conditions for small samples with dimensions 10 x 8 cm2. Application of this treatment as continuous process in industrial conditions will be a great challenge.

Before publication the paper has to be revised.

Comments:

1/ Generally English has to be improved.

There are unfortunate expressions and misused terms in the text.

For example:

  1. 79 “As a contribution to the literature, in the present paper, we studied the functionalization …”. I think the goal of research is to study a new treatment, not to produce next paper.

l.103 the term “lower” is relative. One can discuss if the magnification of 10 000 belongs to low magnifications. Preferably other term should be used.

  1. 111 The term “amplification” does not fit to microscopic images. Commonly used is the term: “magnification”

l.173 To investigate the chemical bonds of the sample. Do you really studied chemical bonds?

There are many grammatical errors in the text.

For example:

  1. 114-115 … was covered with 5 mm thick dielectric made of silicone material. Probably the word “layer” is missing.
  2. 172 … coupled to investigated the chemical modifications

l.179 The source is Al-Ka radiation source ….

  1. 183 … curve method to analyses the data

There are much more unfortunate expressions, misused terms and grammatical errors. All have to be eliminated.

Response: We agree with the reviewer for typos and grammatical errors. The entire text of the article was carefully revised in relation to the English language.

     2. There is lack of information regards fabric in the text. The reference concerning this point is given. Nevertheless, short characteristic of the fabric used in investigations should be presented.

Response: More details on the fabric used were inserted in the first paragraph of topic 2.1.

     3. Results and discussion

The first three sentences in the paragraph (l.190-195) are unnecessary. Two sentences are suitable for the Introduction. One sentence repeats information from the experiment.

Response: We agree with the reviewer, the first three sentences in the paragraph (l.190-195) have been suppressed from the text.

Round 2

Reviewer 1 Report

The authors considered the remarks. The paper is publishable.